# Individuals’ Vulnerability Based Active Surveillance for TB: Experiences from India

**DOI:** 10.3390/tropicalmed7120441

**Published:** 2022-12-15

**Authors:** Shibu K. Balakrishnan, Rakesh P. Suseela, Sunilkumar Mrithyunjayan, Manu E. Mathew, Suresh Varghese, Shubin Chenayil, Suja Aloysius, Twinkle Prabhakaran, Sreenivas A. Nair

**Affiliations:** 1World Health Organization Technical Support Network (NTEP), Thiruvananthapuram 695001, India; 2State TB Elimination Program, Kerala State Health Services Department, Directorate of Health Services, Thiruvananthapuram 695001, India; 3Country and Community Support for Impact, Stop TB Partnership Secretariat, 1218 Geneva, Switzerland

**Keywords:** active case finding, health system efficiency, TB risk factors, vulnerability reduction

## Abstract

Community-based active TB case finding (ACF) has become an essential part of TB elimination efforts in high-burden settings. In settings such as the state of Kerala in India, which has reported an annual decline of 7.5% in the estimated TB incidence since 2015, if ACF is not well targeted, it may end up with a less-than-desired yield, the wastage of scarce resources, and the burdening of health systems. Program managers have recognized the need to optimize resources and workloads, while maximizing the yield, when implementing ACF. We developed and implemented the concept of ‘individuals’-vulnerability-based active surveillance’ as a substitute for the blanket approach for population/geography-based ACF for TB. Weighted scores, based on an estimate of relative risk, were assigned to reflect the TB vulnerabilities of individuals. Vulnerability data for 22,042,168 individuals were available to the primary healthcare team. Individuals with higher cumulative vulnerability scores were targeted for serial ACF from 2019 onwards. In 2018, when a population-based ACF was conducted, the number needed to screen to diagnose one microbiologically confirmed pulmonary TB case was 3772 and the number needed to test to obtain one microbiologically confirmed pulmonary TB case was 112. The corresponding figures in 2019 for individuals’-vulnerability-based ACF were 881 and 39, respectively. Individuals’-vulnerability-based active surveillance is proposed here as a practical solution to improve health system efficiency in settings where the population is relatively stationary, the TB disease burden is low, and the health system is strong.

## 1. Introduction

Closing the gaps in tuberculosis (TB) notification, preventing transmission through early diagnosis, and providing appropriate treatment to prevent mortality are some of the most important objectives of national TB programs (NTPs). Contact screening has been implemented by NTPs as a routine strategy for enhanced case detection. In addition, community-based active TB case finding (ACF) has become an essential part of NTPs in India [1]. A recent systematic review highlighted that the screening of general populations may outperform current case-finding practices and more at-risk groups should be considered for inclusion in screening recommendations [2]. Inhabitants of urban slums, congregations, prisoners, migrants, underserved groups, and people in geographical areas with restricted access are the at-risk groups which are considered key populations for community-based ACF. However, if the process is not well targeted, ACF may end up with a less-than-desired yield, the wastage of resources, and the burdening of health systems [3].

Kerala, the southern Indian state, has made remarkable progress in improving people’s health, as evidenced by a higher life expectancy and a lower infant mortality rate [4]. Its primary healthcare delivery system is well established, with a primary health center (PHC) for every 25,000 people, staffed by a medical doctor. For every 5000 people, there are two multi-purpose health workers and for every 1000 people there is a community health volunteer (an Accredited Social Health Activist = ASHA). Multipurpose health workers (MPHWs) are expected to visit every household in their area at least once every three months. The TB program in the state is well integrated with primary healthcare delivery.

Kerala has experienced a 7.5% annual decline in the incidence of TB since 2015 and was certified for the same by the Government of India under its sub-national TB-free certification process in 2022 [5]. Contact screening and community-based ACF started in the state in 2007 and achieved state-wide coverage in 2010. The total presumptive TB examination rate continued to increase from 888/100,000 people in 2010 to 1442/100,000 in 2019, which was among the highest rates in the country [5,6]. The state reported 87 and 72 TB (new and retreatment, all forms) cases per 100,000 people in 2015 and 2019, respectively [6]. Along with the decline in TB incidence, the yield from community-based ACF started declining since 2015, even after periodic house visits by ASHAs, MPHWs, and NTP staff in the identified key populations and geographical areas such as urban slums, tribal hamlets, coastal areas, difficult-to-reach villages, elderly people’s homes, prisons, and other congregation settings. The number of TB tests overstretched the capacity of most of the TB microscopy and rapid molecular diagnostic centers and X-ray facilities. With the presumptive TB examination rates going up and the yield of the testing coming down, program managers acknowledged the need to optimize their resources and workloads and to maximize the yield. We developed the concept of individuals’-vulnerability-based active surveillance for TB as a substitute for the blanket approach for population/geography-based screening.

## 2. Materials and Methods

### 2.1. Concept

Common risk factors (vulnerabilities) for TB in Kerala were listed. The process undertaken for identifying TB risk factors and estimating their relative risks through a systematic literature review of locally available studies and programmatic data have been described elsewhere [7]. Weighted scores were assigned to the vulnerabilities based on the estimates of relative risk [7,8]. The use of an expert group consensus has been widely accepted as a method for providing a basis for decision-making when there is limited evidence or when there are doubts about the applicability of evidence that has been generated from other populations. An expert group was assembled, consisting of nine purposefully selected experts, including 2 program managers, 2 epidemiologists, 3 public health experts, and 2 clinicians, all of whom had worked for a TB program for more than 5 years. The expert group looked at the relative risks of each vulnerability factor [7], as well as the quality and strength of evidence in the local contexts, and discussed and finalized the final weighted scores for each vulnerability factor. The vulnerability factors considered (with their weighted scores in brackets) were as follows—household contact (5), healthcare worker (3), diabetes (3), tribal (3), mine/quarry worker (2), liver/kidney disease (2), bedridden/palliative care (2), past TB (3), coastal residence (2), chronic lung disease (2), smoking (3), harmful consumption of alcohol (pattern of use that is causing damage to physical or mental health) (2), slum dwelling (1), and migrant (2). HIV was not included since complete information on all people living with HIV (PLHIV) are available through TB and HIV programs and these people are being screened for TB on a monthly basis. Malnutrition was not included due to the practical difficulties involved with health volunteers carrying weighing scales and stadiometers to identify the nutritional status of each individual.

Scores for individual factors were added for individuals based on their associated vulnerability factors. For example, a person who is classified as tribal and who smokes will receive a score of 6. People with cumulative vulnerability scores of 5 or above were considered potential candidates for active screening on a quarterly basis. Persons with cumulative scores from 1 to 5 were considered for intensified case finding when they reported to a health facility for any reason. Persons with a vulnerability score of zero were expected to voluntarily report at the onset of symptoms. These cut-offs were fixed arbitrarily based on consensus through the use of the nominal group technique by the expert group. When fixing the cut-off, the operational feasibility of various cut-offs, such as the efforts required by the health system to actively screen those with cumulative scores of 5 and above vs 3 and above, were also considered.

### 2.2. Process of Vulnerability Score Compilation

A base list of individuals with vulnerabilities was compiled from the family health survey database, the database maintained by the PHC team with details of all individuals residing in the concerned area. A door-to-door campaign was organized with the following objectives—(1) to sensitize every citizen of the state with regards to TB and (2) to update the base list with additional self-reported vulnerabilities. Communication tools and survey tools were developed for the campaign. A pre-structured pilot-tested survey questionnaire with yes/no answers was prepared with accessory formats for multi-level compilation [9]. ASHA, MPHW, and NTP staff were trained for data collection using a standardized module. Sub district/district TB program managers provided training to the staff at the periphery, as well as multi-layer quality control at all stages. The campaign started in January 2018 and had ended by July 2018. Verbal consent was obtained from the participants. Paper-based files were digitalized, compiled, and analyzed using Microsoft Excel for local action. The digitalization of data was completed by November 2018. While line-lists were maintained at local primary health centers as vulnerability databases for future updates, aggregates were compiled at the district and provincial program levels.

### 2.3. Authentication of the Vulnerability Score Cut-Off

We attempted to authenticate the vulnerability score cut-off. All (*n* = 3121) microbiologically confirmed TB cases reported among the surveyed individuals (22,042,168) during the period from 1 January to 30 April 2019 were searched in the vulnerability database to obtain their vulnerability scores. Relative risk scores with 95% confidence intervals were calculated to estimate the risk of developing active TB disease among the moderate- and high-vulnerability groups compared with the low-vulnerability group.

### 2.4. Individuals’-Vulnerability-Based Active Surveillance for TB

Individuals’-vulnerability-based ACF was first conducted from October to December 2019. In the primary health system, health volunteers screened all individuals with a higher vulnerability (scores of 5 or more) for any of the four-symptom complex (4S = cough, fever, weight loss, and night sweats) and testing for TB was performed through rapid molecular tests. Individuals with vulnerability factors such as diabetes mellitus, tobacco use, and chronic respiratory diseases were also linked with vulnerability reduction activities. Though the plan was to conduct the individual-vulnerability-based ACF quarterly, it was delayed due to the COVID-19 pandemic and the next round of ACF occurred only during the period from October to December 2020.

Descriptive statistics using frequencies and percentages were calculated for various indicators of active case finding through a population/geography-based approach in 2018 and individual-vulnerability-based surveillance in 2019 and 2020. The chi-squared test was used to test for differences in the proportions measured in the years 2018 and 2019 and *p* values were calculated.

A flowchart representing the process followed for individual-vulnerability-based active surveillance of TB is shown in Figure 1.

## 3. Results

Of 34.1 million residents, data for 22,042,168 were available at the primary health center level in digital formats. Of these, 81.2% (17,905,748) of the residents did not report any vulnerability, 15.25% (3,360,618) had cumulative scores of 1–4, and 3.52% (775,802) had a score of 5 or above.

Of the 3121 microbiologically confirmed TB cases that occurred between 1 January and 30 April 2019, 824 (26.4%) had no recorded vulnerabilities, 743 (23.8%) had a vulnerability score between 1 and 4, and 1554 (49.8%) of the TB cases had vulnerability scores of 5 or more. We observed that individuals with higher vulnerability scores (≥5) had a higher risk (RR 43.4 (95% CI 39.9–47.2)) of developing TB disease when compared with individuals with no vulnerability. There was also a significant difference between the risks of developing TB among the moderate- and high-vulnerability score groups. The results of the authentication exercise are summarized in Table 1.

In 2018, when population/geography-based ACF was conducted, the number needed to screen (NNS) to diagnose one microbiologically confirmed pulmonary TB case was 3772 and the number needed to test (NNT) to obtain one microbiologically confirmed pulmonary TB case was 112. In 2019, when individuals’-vulnerability-based ACF was conducted, the figures were 881 and 39, respectively. The proportions of presumptive TB cases identified among those screened in 2018 and 2019 were 3.4% and 5.7%, respectively (*p* < 0.001). The test positivity rate was observed to be higher in 2019 (2.6%) compared to 2018 (0.9%), (*p* < 0.01). The proportions and figures for 2020, when ACF was attempted during the COVID-19 pandemic, remained the same as that of 2019. A comparison of the results for population/geography-based ACF and individual-vulnerability-based ACF is presented in Table 2.

## 4. Discussion

The median NNS for India in 2018 was 2080 (interquartile range (IQR) = 517–4068) and in 2019 it was 2468 (IQR = 1050–7924) [10]. The same for the state of Kerala was 3772 in 2018. Upon the implementation of vulnerability-based active surveillance, the NNS dropped to 881 in Kerala. Through the mapping of individuals’ vulnerability, the prevalence was artificially increased in 2019 in the population targeted for active screening, improving the yield of ACF. In 2019, approximately one third of the costly molecular tests were also avoided, in addition to the lowering of the workloads of TB laboratories. In 2020, it was demonstrated that the strategy worked well even during the COVID-19 pandemic. Thus, individual-vulnerability-based TB surveillance could be considered as a solution to improve system efficiency by optimizing resources and efforts in settings where the population is relatively stationary, the TB disease burden is low, the primary healthcare system is strong, and where geography/population-based ACF has limited value.

The mapping of individuals’ vulnerability helped the program managers to identify the individuals who needed to be serially followed-up. It also provided the opportunity for the primary healthcare system to reduce the vulnerability of individuals by linking them to appropriate programs. For example, people with diabetes were linked to diabetes clinics and people with tobacco use were offered tobacco cessation services. Vulnerability-based surveillance and vulnerability reduction could be appropriate interventions for ensuring equity in primary TB care delivery and enhancing convergence in order to address the social and clinical determinants of TB. This may be a good investment, as addressing these vulnerabilities might reduce total healthcare costs for the community in the long run as these are vulnerabilities not only for TB but for many of the most common chronic diseases and conditions. Individuals with high vulnerability scores could also be considered as priority groups when scaling up preventive therapy for TB through a ‘test and treat’ approach at the last mile.

The information obtained from vulnerability mapping is important not only for the elimination of TB, but can also be of use to program managers or policy makers to improve the health of the population. The vulnerability database helped the state during the COVID-19 pandemic to ensure proactive care to vulnerable individuals, including providing counseling and education and delivering medicines for chronic diseases at households, which helped in reducing the overall case fatality due to COVID-19. During the COVID-19 pandemic, when the TB case findings showed a decrease, individual-vulnerability-based ACF helped the state to recover rapidly and find the ‘missing TB cases’ [11].

There are limitations in the implementation of this concept. The vulnerability factors were self-reported; hence, there existed a probability of underreporting. The program managers are working in the field in order to improve the quality of the data and periodically update it, along with routine house-to-house visits by multipurpose health workers as part of the system. Undernutrition, one of the most important vulnerabilities, was not considered when calculating the vulnerability scores. Even with all these limitations on the quality of the vulnerability data, approximately 50% of the newly notified TB cases emerged from 3.5% of the population, who had vulnerability scores of 5 and above. The state would be able to address 50% of their TB cases by focusing on 3.5% of the vulnerable individuals. This percentage may become higher if all vulnerabilities could be captured correctly. Screening was largely 4S-based, reducing the specificity of presumptive TB identification. Paper-based reporting limited the availability of line lists to primary care facilities, leaving the program managers with the option of using aggregates only, and so a sensitivity analysis using various threshold scores could not be attempted. We presented the results of two rounds of individuals’-vulnerability-based ACF, but results obtained from the serial screening of the same population could provide more insights. The perceptions of program managers, primary healthcare providers, and beneficiaries regarding the acceptability, feasibility, and sustainability of the approach, as well as operational challenges related to this approach, need to be captured systematically. Documenting the cost-effectiveness of the approach is also essential. The collateral benefits of this approach in reducing delays in the diagnosis of TB need to be studied and documented. The incorporation of appropriate digital health technologies such as e-health would provide opportunities for the easier mapping of vulnerability factors and active screening through system-generated alerts instead of door-to-door ACF.

## 5. Conclusions

Individual-vulnerability-based active surveillance could provide a practical solution to improve health system efficiency and optimize resources in settings where the population is relatively stationary, the TB disease burden is low, and the primary health system is strong.

## Figures and Tables

**Figure 1 tropicalmed-07-00441-f001:**
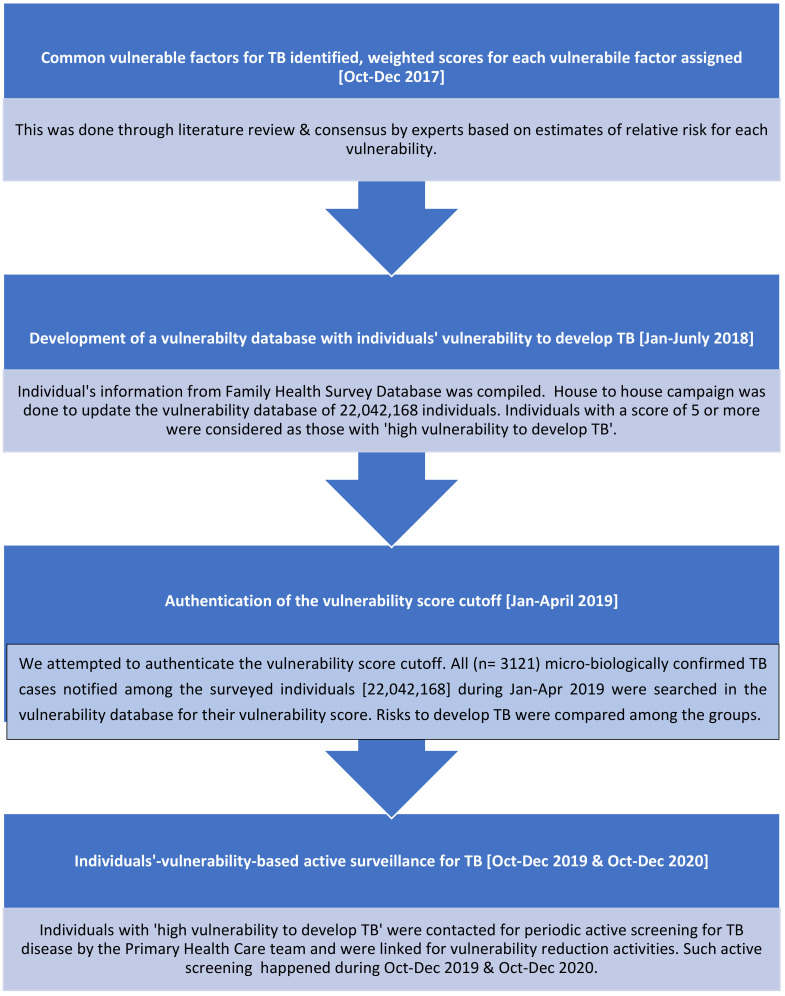
Flowchart representing the process for individual-vulnerability-based active surveillance of TB.

**Table 1 tropicalmed-07-00441-t001:** Summary of TB cases that emerged from various vulnerability groups based on vulnerability scores.

Vulnerability Groups	Total Number of Individuals (a) (Mapped during Jan–July 2018)	Out of (a), Number Developed Microbiologically Confirmed TB Disease Cases (Routine Program Data Jan–April 2019)	RR (95% CI)
Low vulnerability to the development of TB (score of 0)	17,905,748 (81.2%)	824 (26.4%)	1
Moderate vulnerability to the development of TB (scores 1–4)	3,360,618 (15.3%)	743 (23.8%)	4.8 (4.3–5.3)
High vulnerability to the development of TB (scores ≥ 5)	775,802 (3.5%)	1554 (49.8%)	43.4 (39.9–47.2)
Total	22,042,168	3121	

**Table 2 tropicalmed-07-00441-t002:** Comparison of yields from population-based ACF during 2018 and individuals’-vulnerability-based ACF during 2019 and 2020.

	Population-Based ACF in 2018	ACF among Individuals with High Vulnerability * to the Development of TB in 2019	ACF among Individuals with High Vulnerability * to the Development of TB in 2020
A. Total population of the state	34,411,267	34,545,868	34,678,294
B. Total individuals mapped for screening for TB	1,600,000	775,802	976,147
C. Out of (B), total individuals screened	1,139,200 (71.2%)	537,371 (69.3%)	547,262 (56%)
D. Out of (C), total presumptive pulmonary TB cases identified	46,707 (4.1%)	30,900 (5.7%)	28,457 (5.2%)
E. Out of (D), total tested for TB	34,096 (73%)	23,585 (76.3%)	21,058 (74%)
F. Out of E, total microbiologically confirmed pulmonary TB case diagnosed	302 (0.9%)	610 (2.6%)	558 (2.6%)
G. Number needed to test to detect one microbiologically confirmed pulmonary TB case (NNT)	112	39	38
H. Number needed to screen to diagnose one microbiologically confirmed pulmonary TB case (NNS)	3772	881	980
I. Microbiologically confirmed pulmonary TB cases diagnosed through ACF as a proportion of total microbiologically confirmed pulmonary TB cases in the state during the concerned period	7.1%	18.3%	16.2%

* High vulnerability indicates a cumulative vulnerability score of 5 or more.

## Data Availability

The data presented in this study are available on request from the corresponding author. The data are not publicly available as they were not collected for research purposes. Restrictions apply to the availability of individual-level vulnerability data. Data are available with the concerned primary healthcare team of Kerala State Health Services and can be obtained with the permission of the Director of Health Services, Kerala State Health Services.

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
