# Peer review of "Individuals’ Vulnerability Based Active Surveillance for TB: Experiences from India"

_tropicalmed, 2022, doi:10.3390/tropicalmed7120441_

Round 1

Reviewer 1 Report

Mostly minor edits and clarification.  See attached file

Author Response

We thank the reviewer for the comments. It was very useful. We believe that addressing those comments have improved the quality of manuscript. All comments by the reviewer were addressed.  Changes done as follows

  1. Removed the word 'upfront' in line 116.
  2.  Added 'verbal consent was obtained from the participants' in line 104

Apart from these, we have done all the language edits suggested by the reviewer.

Reviewer 2 Report

Review for ‘Individuals’ vulnerability based active surveillance for TB: Experiences from India’

The authors have conducted a study which addresses the very important topic of how to optimized community-based ACF in a setting with estimated high TB burden and also limited resources to provide good coverage to all. Novel methods like this are needed to prioritize areas for active case finding in community and facility-based settings. The authors should be commended for doing this work and writing this manuscript. The discussion nicely addresses how the results of this study are relevant for public health planning both for TB and beyond the realm of TB.

There are a number of important issues which should still be addressed before publication, listed below.

Author Response

We thank the reviewer for all the comments. We really appreciate the efforts taken by the reviewer for reviewing the manuscript. We found all the comments relevant and important.  We felt that addressing those comments have immensely improved the quality of our manuscript. Once again, we take the opportunity to thank the reviewer for all the guidance and comments.

Reviewer 3 Report

The active case finding program is very interesting and is worth reporting. I think this report is very useful. However, I would like to know a little more.

TB diagnosed through ACF as a proportion of total microbiologically confirmed pulmonary TB in the state during the concerned period (October to December, 3 months) was 16 to18%. Therefore, in comparison to one year case detection, the proportion is around 4%. Is my understanding right? If so, does this small proportion have impact on the source reduction of TB infection? 

Individualized active case finding has difficulty of access to the individual with TB risk. Economical analysis (or cost effectiveness analysis) will be further useful for the implementation and I expect the CEA to be done in future.

Author Response

We thank the reviewer for the comments.  

Comment 1: TB diagnosed through ACF as a proportion of total microbiologically confirmed pulmonary TB in the state during the concerned period (October to December, 3 months) was 16 to18%. Therefore, in comparison to one-year case detection, the proportion is around 4%. Is my understanding right? If so, does this small proportion have impact on the source reduction of TB infection? 

Author's reply: 

The proportion of cases obtained through individual’s vulnerability mapping was double when compared to the population/geography based ACF. Conceptually, we expect that we may get 16-18% whenever we do ACF (every quarter) for some time. Individual’s vulnerability based active surveillance may cut down the delay in diagnosis of the TB cases (not only this 16-18%), as 50% of TB cases emerge from this 3.5% population and these individuals with vulnerability will be contacted periodically. We added the need to document the impact on delays in line 203-204 in discussion. The approach also provides an opportunity to reduce vulnerability of those individuals and to offer TB Preventive Therapy- We have included these in discussion- lines 166-186.

Comment 2: Individualized active case finding has difficulty of access to the individual with TB risk. Economic analysis (or cost effectiveness analysis) will be further useful for the implementation and I expect the CEA to be done in future.

Author's Reply: We agree with the reviewer. Individualised active case finding has value in particular settings. We mentioned in discussion as   “ individuals’ vulnerability based TB surveillance could be considered as a solution to improve the system efficiency by optimising the resources and efforts in setting where population is captive, TB disease burden is low and primary health care system is strong and where geography/population based ACF has limited value.” Line 161-165. Same has been highlighted in conclusion. In line 203, in discussion we mentioned the need to perform economic analysis.

Round 2

Reviewer 2 Report

Please find my feedback to the updated manuscript draft attached. 

Author Response

Point by point Response attached as word file. 
